# Female Sex Determination Factors in *Ceratitis capitata*: Molecular and Structural Basis of TRA and TRA2 Recognition

**DOI:** 10.3390/insects14070605

**Published:** 2023-07-04

**Authors:** Maryanna Martina Perrotta, Francesca Lucibelli, Sarah Maria Mazzucchiello, Nicole Fucci, Bruno Hay Mele, Ennio Giordano, Marco Salvemini, Alessia Ruggiero, Luigi Vitagliano, Serena Aceto, Giuseppe Saccone

**Affiliations:** 1Department of Biology, University of Naples “Federico II”, 80126 Napoli, Italy; 2Institute of Biostructures and Bioimaging (IBB), CNR, 80131 Napoli, Italy

**Keywords:** sex determination, development, alternative splicing, autoregulation, protein–protein interactions, yeast two-hybrid, alpha fold, structure, gene regulation, pest control

## Abstract

**Simple Summary:**

In insects, sex determination is generated using intricate and variegate biological processes that may be effectively described as variations on a common theme. In the model system *Drosophila melanogaster*, genetic and biochemical studies have shown that the female-specific Transformer (TRA) and the non-sex-specific Transformer2 (TRA2) are RNA-binding proteins that physically interact to promote female differentiation by female-specific alternative splicing of downstream genes. This *tra* gene responds and transduces different primary sex-determining signals, and its master function is widely conserved in Diptera, Coleoptera, and Hymenoptera. Here, combining yeast two-hybrid and computational methodologies, we demonstrate that the TRA and TRA2 orthologs of the agricultural pest *Ceratitis capitata* physically interact through a molecular mechanism that could be evolutionarily conserved in other species. These technical approaches can be helpful to verify or to identify other proteins interacting with TRA and TRA2, for example, those promoting male sex determination in this and other species, as well as to design new compounds that could induce masculinization of XX individuals in applications of the Sterile Insect Technique.

**Abstract:**

In the model system for genetics, *Drosophila melanogaster*, sexual differentiation and male courtship behavior are controlled by sex-specific splicing of *doublesex* (*dsx*) and *fruitless* (*fru*). In vitro and in vivo studies showed that female-specific Transformer (TRA) and the non-sex-specific Transformer 2 (TRA2) splicing factors interact, forming a complex promoting *dsx* and *fru* female-specific splicing. TRA/TRA2 complex binds to 13 nt long sequence repeats in their pre-mRNAs. In the Mediterranean fruitfly *Ceratitis capitata* (Medfly), a major agricultural pest, which shares with Drosophila a ~120 million years old ancestor, *Cctra* and *Cctra2* genes seem to promote female-specific splicing of *Ccdsx* and *Ccfru*, which contain conserved TRA/TRA2 binding repeats. Unlike Drosophila *tra*, *Cctra* autoregulates its female-specific splicing through these putative regulatory repeats. Here, a yeast two-hybrid assay shows that CcTRA interacts with CcTRA2, despite its high amino acid divergence compared to Drosophila TRA. Interestingly, CcTRA2 interacts with itself, as also observed for Drosophila TRA2. We also generated a three-dimensional model of the complex formed by CcTRA and CcTRA2 using predictive approaches based on Artificial Intelligence. This structure also identified an evolutionary and highly conserved putative TRA2 recognition motif in the TRA sequence. The Y2H approach, combined with powerful predictive tools of three-dimensional protein structures, could use helpful also in this and other insect species to understand the potential links between different upstream proteins acting as primary sex-determining signals and the conserved TRA and TRA2 transducers.

## 1. Introduction

As in all other pluricellular living forms, developmental choices in insects are often controlled by differential transcription of master genes and their gene targets. On the contrary, in several investigated insect species belonging to Diptera, Hymenoptera, and Coleoptera, the sex-determination regulatory pathway is based mainly on the alternative splicing of a master gene, *transformer* (*tra*), which is switched on or off in female and male sex, respectively [1,2,3,4,5,6]. In the female sex, *tra* female-specific transcripts encode for functional TRA, a Serine/Arginine-rich disordered protein (RS-type). In contrast, the male counterpart includes an exonic sequence containing a stop codon and prematurely truncates the *tra* open reading frame. This genetic pathway has both flexibility and robustness. The flexibility relies on the diverging upstream sex-determining primary signals observed in insect species, families, and orders [7,8,9,10]. Its robustness relies on conserved splicing factors responsible for regulating female-specific splicing, hence, determining and maintaining the female sex determination [9,11,12,13].

In the *Drosophila melanogaster* model system and related Drosophilidae, during embryogenesis, two doses of the X chromosome compose the primary signal activating the master gene, *Sex-lethal* (*Sxl*). At the same time, XY embryos follow a default male sex determination [14,15,16,17,18]. *Sxl* encodes a female-specific splicing regulator that promotes female-specific splicing of *tra* and of *Sxl* itself pre-mRNA splicing, maintaining female sex determination during all development [19]. Structural and functional analysis of the *Drosophila melanogaster transformer* (*tra*) and *transformer-2* genes, which encode these two factors, revealed their involvement in female sex determination by promoting female-specific splicing of downstream gene regulators, including *doublesex* (*dsx*) and *fruitless* (*fru*), responsible for modulating hundreds of sexual differentiation genes [6,20,21,22,23,24,25]. The Drosophila *dsx* and *fru* female-specific exons share copies of a 13 nt long cisregulatory element acting as a splicing enhancer (*dsx*RE, *dsx* repeat element) and recognized by the TRA/TRA2 interacting complex [26,27,28,29,30,31,32]. UV-crosslinking and purified recombinant proteins showed that TRA and TRA2 bind to this splicing enhancer [33]. Coimmunoprecipitation using a Drosophila cell line also detected TRA and TRA2 protein interaction. A yeast protein–protein interaction assay (yeast two-hybrid Y2H; [34]) showed that TRA2 interacts with itself and with TRA also in the absence of *dsx* pre-mRNA [35]. 

Based on these Drosophila data, by inference, a similar model of action was proposed for TRA and TRA2 orthologous proteins found conserved in the Mediterranean fruitfly *Ceratitis capitata* [36,37,38] and in many other insect species [6]. Indirect evidence supporting this evolutionary conservation of the TRA-TRA2 interaction model: (1) A conserved female-specific alternative splicing of *tra*, *dsx,* and *fru*; (2) A conserved *tra* on/off splicing regulation in the two sexes; (3) Conserved TRA/TRA2 binding sites in female-specific exons of *dsx* and *fru* orthologues [6,39,40,41,42,43]; (4) The ability of a non-Drosophilidae TRA (*Ceratitis capitata* TRA protein), when expressed in Drosophila transgenic mutant flies, to rescue endogenous *tra* function, only in the presence of a Drosophila *tra2* endogenous gene, despite its very low protein length (CcTRA/DmTRA, 429aa/197aa) and sequence conservation (CcTRA/DmTRA 35% similarity and 24% identity) [44].

Nevertheless, in *Ceratitis capitata*, in addition to the evolutionary conservation of the *tra*/*tra2* > *dsx*/*fru* genetic developmental module, there are relevant differences: the Y chromosome dictates male sex determination on a default female sex determination [45,46]; the *Ceratitis Sxl* orthologue is not involved in sex determination [47]. Moreover, Tra/Tra2 putative binding elements are unexpectedly present within the *Cctra* male-specific exons region, a finding that supported the conclusion that the Ceratitis *tra* gene has a novel autoregulatory function [36]. A maternal input of *tra* and *tra2* genes is essential to start the positive autoregulatory *Cctra* loop and act as a feminizing primary signal [36,38]. The *Ceratitis tra* autoregulation was found widely conserved in many different insect species, including *Musca domestica* [48,49], but not in Drosophila, which lost it before or during the emergence of the Drosophilidae family [6,50,51]. 

In Ceratitis XY embryos, the Y-linked male-determining *MoY* gene encodes a novel short protein, which induces either direct or indirect male-specific unproductive splicing of *Cctra* (within two hours in 5–7 h old embryos), likely inhibiting the actions of maternal CcTRA and CcTRA2 on *Cctra* zygotic pre-mRNAs [52]. Similarly, as in Drosophila, the male-specific *Cctra* mRNAs include male-specific exons introducing premature stop codons and, thus, encoding shorter, likely nonfunctional, proteins (CcTRAM1 59 aa and CcTRAM2 99 aa long) [36]. The continuous reduction/absence of CcTRA protein in the following embryogenic stages of XY individuals likely leads to the collapse of the *Cctra* positive feedback loop and to permanent *Cctra* male-specific splicing. 

In support of this model, temporary depletion of *Cctra* or *Cctra2* mRNAs during embryogenesis of XX by RNAi or dCas9 led to a permanent shift of *Cctra* splicing into the male-specific pattern (mimicking the effect of *MoY* in XY embryos) and to the development of XX males [36,38,53]. Unlike *Drosophila* XX *tra/tra* males, the reverted *Ceratitis* XX males are fertile, indicating the lack of relevant Y-linked fertility factors. 

In this intricate framework, we experimentally assessed the direct binding of the *C. capitata* TRA and TRA2. We also evaluated the possibility that MOY explicates its function by interacting with CcTRA or CcTRA2. Moreover, using recently released machine-learning tools that effectively predict protein structures and their complexes, we also generated an atomic-level model of the CcTRA/CcTRA2 complex. 

## 2. Materials and Methods

### 2.1. Rearing of Ceratitis capitata

We used the Benakeion strain, developed by P. A. Mourikis (Benakeion Institute of Phytopathology, Athens, Greece). The strain was reared in laboratory conditions at 26 °C, 60% relative humidity, and exposed to 12 h/12 h light–dark cycles. Adults were fed with a mixture of sugar and yeast powder (3:1). The larval food was made with 30 gr paper, 400 mL dH2O, 2 mL HCl (Ci = 2%), 10 mL cholesterol (Ci = 2.5%), 8 mL benzoic acid (Ci = 4%, pH = 2.8), 30 gr yeast powder, and 30 gr sugar. Eggs were collected into Petri dishes filled with larval food. Pupae were collected and stored in Petri dishes until eclosion.

### 2.2. RNA Isolation and cDNA Synthesis

According to manufacturer instructions, total RNA was extracted from male and female adult flies using TRIzol (Ambion, Austin, TX 78744, USA). After the extraction and quantification, 1 mg of total RNA was reverse-transcribed using LunaScript RT SuperMix Kit (New England Biolabs, Ipswich, MA 01938, USA) and oligo-dT reverse transcriptase-based protocol. 

### 2.3. Yeast Two-Hybrid Assay

A Y2H assay based on the GAL-4 system (Matchmaker, two-hybrid system; Clontech, Palo Alto, CA 94303-4230, USA) was used to investigate the interaction between the MOY, CcTRA, and CcTRA2 proteins. The full-length coding regions of *MoY* (MK165756.1), *Cctra* (AF434936.1), and *Cctra2* (NM_001279408.1) were amplified by PCR using the DreamTaq polymerase (Invitrogen-ThermoFisher, Waltham, MA 02451, USA) based on the manufacturer instructions, using the primer pairs listed in Table 1.

The *MoY-*, *Cctra-,* and *Cctra2*-amplified full-length coding regions were cloned into the bait vector pGBT9 (Clontech) containing the yeast Gal4 DNA binding domain (BD) and the prey vector pGAD424 (Clontech) containing the yeast Gal4 activation domain (AD). In brief, the PCR-amplified products and the vectors were digested using the restriction endonucleases *Eco*RI and *Sal*I (Promega, Fitchburg, WI 53711, USA). The manufacturer’s instructions performed the ligase reaction using the T4 DNA ligase enzyme (NEB). Sequence analysis confirmed that the DNA fragments cloned into pGBT9 and pGAD424 were in frame with the Gal4 BD and AD, respectively. 

The *Saccharomyces cerevisiae* strain AH109 was transformed with all the prey and bait recombinant vector combinations using the LiAC/SSDNA/PEG transformation method [54]. The experiment was conducted in triplicate. The double-transformed cells were plated on a Synthetic-Defined (SD) agar medium that lacked leucine and tryptophan (SD/-Leu/-Trp) and incubated at 30 °C for 3–4 days to verify the presence of the plasmids. The positive colonies were transferred onto a selective SD medium lacking tryptophan, leucine, and histidine (SD/-His/-Leu/-Trp) in 20 mM of 3-amino triazole (3-AT).

The protein–protein interaction is verified by the transcriptional activation of the reporter gene *HIS3* and the consequent growth of colonies.

In addition, the self-activation of the proteins was tested by the single transformation of yeast cells with the recombinant pGBT9 vectors (BD) and growth in SD medium without histidine and tryptophan (SD/-His/-Trp) containing 20 mM 3-AT. As negative controls, empty pGBT9 and pGAD424 vectors were used in double transformation experiments with the recombinant vectors.

### 2.4. Structural Predictions

Three-dimensional structures of TRA and TRA2 proteins were predicted using the AlphaFold (AF) v2.0 algorithm [55,56], as implemented in the Colab server (https://colab.research.google.com/github/sokrypton/ColabFold/blob/main/AlphaFold2.ipynb) (accessed on 25 November 2022) [57]. 

Predictions were performed without considering any experimental structural template derived from homologous proteins (template_mode: none) and using the maximum allowed number (i.e., 48) of recycles. The best-predicted model (rank 1) out of the five computed by AF was considered and analyzed after that. The reliability of the AF predictions was assessed by evaluating the Local Distance Difference Test (LDDT) score and the Predicted Aligned Error (PAE) [55,56,57]. The LDDT indicator is a *per-residue* confidence score. Protein regions showing values of LDDT higher than 70 are expected to be modeled with reasonable accuracy. 

On the contrary, residues showing LDDT values lower than 50 likely correspond to regions that do not adopt single structured states in physiological conditions that may become structured when involved in biomolecular partnerships. PAE matrices report the estimated errors in the relative position of pairs of residues of protein–protein complexes. Low PAE values of pairs of residues belonging to different proteins within the complex suggest that the prediction of their relative positions and orientations is reliable.

## 3. Results

### 3.1. CcTRA2 Interacts with CcTRA2 Itself and with CcTRA

To experimentally assess the direct interaction between CcTRA and CcTRA2, we conducted yeast two-hybrid (Y2H) assays in which each of the two proteins acted either as bait or prey. To this aim, we amplified the corresponding cDNA fragments by RT-PCR using RNA extracted from adult medfly females, encoding CcTRA and CcTRA2. Each cDNA product was cloned in the bait pGBT9 vector (containing the Gal4 Binding Domain, BD) and in the prey pGAD424 vector (containing the Gal4 DNA Activation Domain, AD).

We verified that the CcTRA-BD and CcTRA2-BD baits do not autonomously activate the reporter gene in the yeast cells without a prey protein. The two constructs were transformed individually with the empty AD reporter vector, and no bait autoactivation was observed (Figure 1, rows 1–2). As a control, we also individually transformed the preys CcTRA-AD and CcTRA2-AD, each with the empty BD bait vector, and no activation was observed (Figure 1, rows 3–4). As a negative control, we confirmed the absence of activation following the empty BD and AD vectors’ cotransformation, indicating the lack of intrinsic transcriptional activity on the GAL4-responsive promoter (Figure 1, row 5).

When we used CcTRA-BD as bait, no interaction was observed with CcTRA-AD (Figure 1, row 6). Interestingly, the cotransformation of the bait CcTRA-BD with CcTRA2-AD as preys showed activation at various dilutions (Figure 1, row 7). In contrast, the cotransformation of CcTRA2-BD with CcTRA-AD showed no interaction (Figure 1, row 8). A similar lack of interactions in reciprocal combinations has been reported in other studies [58,59,60,61]. This result is possible because steric constraints or improper folding can occur in specific recombinant protein/GAL4 domain combinations.

In addition, we detected self-interaction of CcTRA2 (CcTRA2-BD cotransformed with CcTRA2-AD, Figure 1, row 9) also at a tenfold lower dilution (1:100) compared to its interaction with CcTRA (1:10). These data suggest that, as in *Drosophila*, *Ceratitis capitata* the orthologous female-determining CcTRA protein exerts its function by interacting with the orthologous CcTRA2 protein partner, to induce female-specific splicing of target pre-mRNAs (*Ccdsx* and *Ccfru*) bearing specific cisregulatory sequences (Tra/Tra2 binding sites).

### 3.2. MOY Does Not Interact with CcTRA/CcTRA2 Proteins in the Yeast Two-Hybrid Assay

Once established through Y2H experiments, a physical interaction between CcTRA/CcTRA2, we applied this approach to corroborate or confute the hypothesis that MOY male-determining protein induces male-specific splicing of *Cctra* by interacting with either CcTRA or CcTRA2 or both. We amplified by PCR a genomic *MoY* fragment containing the ORF (*MoY* is an intronless gene) from genomic DNA extracted from adult males and cloned it in the vectors BD and AD. No MOY-BD bait autoactivation was observed, as the MOY-BD does not autonomously activate the reporter gene in the yeast cells without a prey protein (Figure 2, row 1). The control of the prey MoY-AD and the empty BD bait vector also showed no activation (Figure 2, row 2). No self-interaction of MOY was observed (Figure 2, row 3). Neither MOY as bait (MOY-BD) nor MOY as prey (MOY-AD) interacted with CcTRA or CcTRA2 (Figure 2, rows 4–7). 

### 3.3. Structural Insights on CcTRA and CcTRA2 Proteins and Their Interactions Unraveled Using Machine-Learning Predictive Approaches

The impressive success of machine-learning approaches in predicting three-dimensional protein structures and their complexes [55,56] suggested the application of this methodology to the CcTRA and CcTRA2 proteins, for which structural data still need to be included. 

In this scenario, we performed trials aimed at predicting the three-dimensional structures of the individual proteins (CcTRA and CcTRA2) as well as their complex by exploiting the abilities of the machine-learning algorithms implemented in AlphaFold (AF) in predicting protein structures starting from their sequences [55,56,57] (see also Methods for details).

#### 3.3.1. Predicted Structural Properties of the Individual CcTRA and CcTRA2 Proteins

The inspection of the CcTRA sequence indicates that it is characterized by the recurrent presence of the Arg-Ser (RS) motif [11,36,62] (Appendix A). Moreover, the peculiar abundance of Arg residues (68 out of 429 residues), distributed along the entire polypeptide chain, is expected to prevent the folding of the protein in a stable structural state due to the excess of positive charges. As for CcTRA, the sequence of CcTRA2 is also characterized by the presence of two RS motifs located in the protein’s N- and C-terminal regions, which are separated by a central RNA recognition domain (RRM—residues ~95–180) [11,62]. This domain is well preserved during evolution. Indeed, the CcTRA2 RRM domain presents a high sequence identity (52%) with that present in human Tra2β. 

Applying the AlphaFold approach (AF) on CcTRA did not yield a stable and reliable three-dimensional model of the protein, in line with the known intrinsically disordered nature. Indeed, except for short protein regions, most of the residues of the predicted model present LDDT values (see Methods for the definition) lower than 50, indicating that these regions of the proteins are essentially unstructured. Again, in line with the expectations, AF runs using the CcTRA2 sequence confirmed the tendency of the protein’s N- and C-terminal regions to be intrinsically disordered. On the other hand, a well-folded domain is predicted in the central portion of the TRA2 sequence. Indeed, residues 100–183 of this predicted model present LDDT larger than 70. The reliability of the model, whose prediction was performed without considering any experimental template, is also corroborated by its similarity to the crystallographic structure of the corresponding domain of human TRA2β [63], as highlighted by the superimposition of the CcTRA2 and TRA2β RRM domain (Figure 3).

#### 3.3.2. Predicted Structure of the CcTRA and CcTRA2 Complex 

Once we assessed the ability of AlphaFold to recapitulate the expected structural properties of CcTRA and CcTRA2, we attempted to predict the structure of the CcTRA–CcTRA2 complex. As shown in Appendix A, in the best-ranked model of the complex, a region with low expected errors in the PAE matrix (see Methods for the definition) of distances between residues of the two proteins can be detected. In particular, the analysis of this blue intermolecular region in the PAE map indicates that it corresponds to the RRM domain of CcTRA2 (residues 100–180) and the residues 55–64 of CcTRA (Appendix A). It is important to note that the region 55–64 of CcTRA also presents relatively high LDDT values (>70), indicating the per-residue reliability of the model (see Methods for details). The inspection of the related three-dimensional model suggests that 55–64 of CcTRA interacts with the exposed β-strand (residues 131–138) of the β-sheet of the RRM domain of CcTra2 (Figure 4). This prediction agrees well with the general stickiness of exposed strands to have a strong tendency to be involved in edge-to-edge associations [64]. In this CcTRA–CcTRA2 complex, residues 60–64 of CcTRA adhere to the C-terminal portion (residues 135–139) of the exposed strand of CcTRA2 (Figure 4A), leading to a β-strand addition in an antiparallel fashion. This protein–protein interaction is stabilized by a remarkable number of hydrogen bonds formed by the exposed hydrogen donors and acceptors present on the exposed strand of CcTRA2 or the main chain (residues 60, 62, and 64) of the side chain (Asn64) of the CcTRA fragment (Figure 4B). 

It is important to note that AlphaFold prediction studies of CcTRA2 with a CcTRA variant in which residues 54–64 were deleted did not yield favorable interactions between the two proteins, as indicated by the absence of intermolecular contact regions with low estimated errors in the PAE matrix (Appendix A). This finding corroborates the role of the CcTRA region 54–64 in the recognition of CcTRA2.

#### 3.3.3. Predicted Structure of the CcTRA and DmTRA2 Complex 

Since previous experiments have demonstrated that CcTRA is also functional in *D. melanogaster* despite the evolutionary distance of these two organisms [44], we also evaluated the basis of its interaction with DmTRA2. A preliminary prediction of the DmTRA2 structure indicates its close similarity to the structure of CcTRA2 (Appendix A), in line with the high sequence identity exhibited by the two proteins. As CcTRA2, a solvent-exposed β-strand (residues 124–132), is also present in DmTRA2. The prediction of the structure of the complex between CcTRA and DmTRA2 indicates that the formation of this complex relies on the same structural determinants that stabilize the CcTRA-CcTRA2 adduct. Indeed, in this case, residues 60–64 of CcTRA adhere to the C-terminal portion (residues 127–131) of the exposed strand of DmTRA2 (Figure 5). Although this finding was somehow expected based on the CcTRA2/DmTRA2 similarity, it nevertheless indicates the prediction’s robustness. 

Although present predictions require some experimental validations, they represent a solid base for identifying the structural basis of TRA-TRA2 recognition in *C. capitata*.

## 4. Discussion

The data presented here demonstrate that the proteins CcTRA and CcTRA2 physically interact. Although this finding could be inferred from the analogy of the known TRA/TRA2 interaction described in *D. melanogaster*, the experimental validation here reported is not obvious considering the remarkable molecular differences between the DmTRA and CcTRA. Indeed, these two proteins do not present any significant sequence similarity (CcTRA/DmTRA 35% similarity by BLOSUM45 with no adjustment) [44] and are characterized by radically different sizes, with CcTRA being much larger than DmTRA (429 versus 197 residues). In this regard, it is also worth noting that even the function of these two proteins is only partially overlapping, as CcTRA, in contrast to DmTRA, can autoregulate its splicing and expression. Therefore, despite the divergence in the evolution of the sequences of CcTRA and DmTRA, their partnership with TRA2 is conserved in these two distantly related species whose common ancestor dates back more than 100 million years ago. 

In our Y2H experiment, the interaction between CcTRA2 and CcTRA is observed when CcTRA2 is fused with the GAL4 AD and CcTRA with the BD, but not in the reciprocal combination. Steric constraints or improper folding can occur in specific recombinant protein/GAL4 domain combinations. This issue has been previously reported when applying the Y2H analysis, and some examples of the lack of interaction between two interactors in both mutual directions can be found in other studies [58,59,60,61]. 

Taking advantage of the recently developed machine-learning methodologies that can provide reliable three-dimensional structures of proteins [55,56,57], we generated a putative model for the CcTRA/CcTRA2 complex. This model suggests that an exposed β-strand of the RRM binding domain of TRA2 anchors the segment 55–64 of CcTRA, which contains both charged and hydrophobic residues. Interestingly, the sequence (LFQRDDIVVN) of this putative TRA2-binding motif of CcTRA is fully conserved in TRA sequences of other Tephritidae and is remarkably well-preserved in other dipteran species such as *Musca domestica* (7 conserved residues out of 10). Surprisingly, it is conserved even in the TRA sequences of distant organisms such as the coleopterans *Onthophagus taurus* (8 conserved residues) and *Trypoxylus dichotomous* (7 conserved residues) despite the extreme evolutionary variability of the sequence of the protein. Modeling data also indicate that this CcTRA motif can interact with the Drosophila TRA2 ortholog, thus providing a structural explanation of the observed ability of CcTRA to rescue female differentiation by inducing female-specific splicing of *Dmdsx* and *Dmfru* in Drosophila transgenic XX [44].

In contrast to *D. melanogaster*, the masculinization process of *C. capitata* relies on a recently identified male-determining factor [52]. This gene (*MoY*), which can suppress the *Cctra* female-specific splicing early during embryogenesis, encodes for a small protein containing 70 residues whose sequence does not present any detectable similarity with functionally and structurally characterized proteins. Therefore, the molecular mechanism underlying MOY’s ability to induce masculinization is unknown. As the Y2H assays reported here successfully unraveled the physical interaction between CcTRA and CcTRA2, we applied this approach to detect possible interactions between MOY and CcTRA/CcTRA2. Data presented here indicate that MOY does not establish direct interaction with either of these proteins. Although the following should be noted: (1) In Y2H assays, false negatives can occur; (2) Species-specific posttranslational modifications may affect protein–protein interactions. These observations indicate that MOY indirectly affects the splicing of CcTRA through a mechanism yet to be uncovered. 

## 5. Conclusions

Collectively, the data presented here show that the Y2H approach, combined with powerful predictive tools of three-dimensional protein structures, could be helpful also in other insect species to understand the potential links between different proteins acting as primary sex-determining signals and the conserved TRA and TRA2 transducers. Moreover, the atomic-level characterization of the CcTRA/CcTRA2 interactions will be helpful for the design and the development of new compounds that, being able to modulate them, could be useful to induce masculinization of XX individuals in applications of the Sterile Insect Technique for agricultural pest insects such as *C. capitata* [65,66,67,68].

## Figures and Tables

**Figure 1 insects-14-00605-f001:**
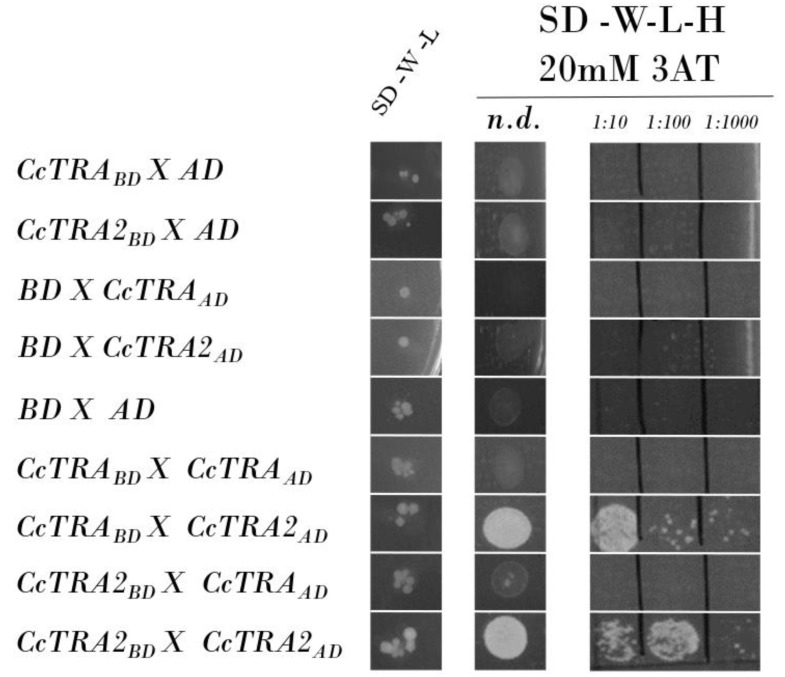
Yeast two-hybrid assay of CcTRA and CcTRA2 interactions. Row 7 shows the interactions of CcTRA and CcTRA2. Row 9 shows the interaction of CcTRA2 with CcTRA2 itself. SD-W-L, Synthetic-Defined agar medium lacking tryptophan and leucine; SD-W-L-H, Synthetic-Defined agar medium lacking tryptophan, leucine, and histidine. n.d., not-diluted yeast liquid culture (10 µL); 1:10, 1:100, and 1:1000 are the dilutions of the yeast liquid culture.

**Figure 2 insects-14-00605-f002:**
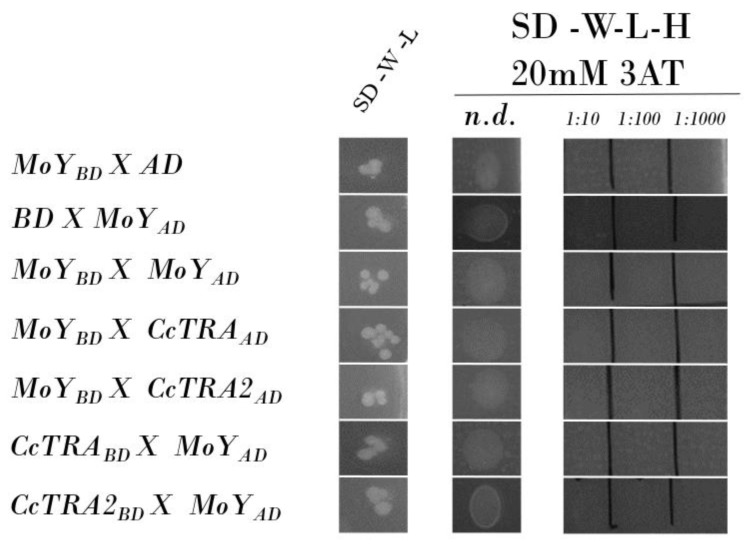
Yeast two-hybrid assay of MOY potential interactions. SD-W-L, Synthetic-Defined agar medium lacking tryptophan and leucine; SD-W-L-H, Synthetic-Defined agar medium lacking tryptophan, leucine, and histidine. n.d., not diluted yeast liquid culture (10 µL); 1:10, 1:100, and 1:1000 are the dilutions of the yeast liquid culture.

**Figure 3 insects-14-00605-f003:**
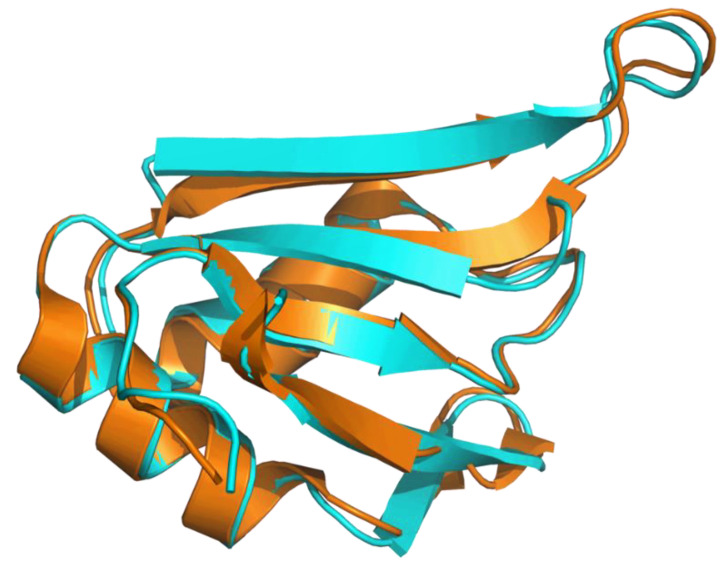
Superimposition of the AF predicted model of the RRM domain of CcTRA2 (cyan) with the experimental model of the human TRA2β RRM domain (orange) (Protein Data Bank code 2RRB).

**Figure 4 insects-14-00605-f004:**
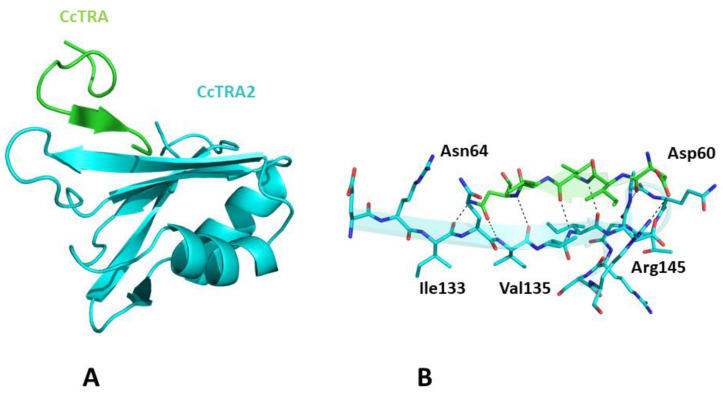
**(A**) Three-dimensional structure of CcTRA (green) and CcTRA2 (cyan) complex. (**B**) The contact region of the complex is highlighted. Hydrogen bonds are also shown.

**Figure 5 insects-14-00605-f005:**
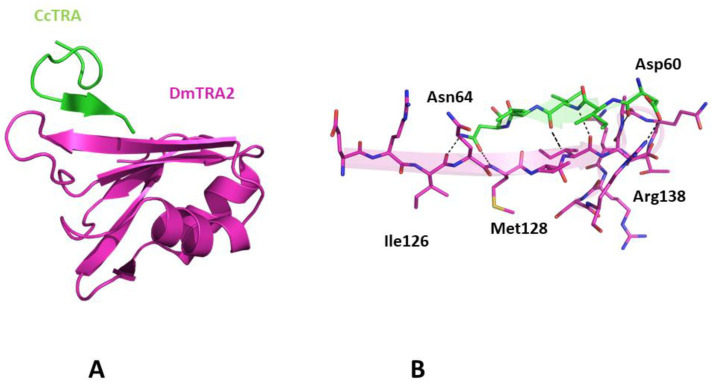
(**A**) Three-dimensional structure of CcTRA (green) and DmTRA2 (magenta) complex. (**B**) The contact region of the complex is highlighted. Hydrogen bonds are also shown.

**Table 1 insects-14-00605-t001:** List of the primers used to amplify target cDNAs.

Primer	Sequence	cDNA bp
Fw_*MoY*_*EcoRI*	CCGGAATTCCGGATGGATATTGGAAATATTTCATCG	352 bp
Rev_*MoY*_*SalI*	AAGTCGACCAATCTGCTAGCATGTGTTCC	
Fw_CcTRA_*EcoRI*	CCGGAATTCCGGATGAACATGAATATTACAAAGGCTTC	1290 bp
Rev_CcTRA_*SalI*	AAGTCGACCTATTTGTGTGTTTTTGGGCGAAA	
Fw_CcTRA2_*EcoRI*	CCGGAATTCCGGATGAGTCCACGTTCACGTAGCC	756 bp
Rev_CcTRA2_*SalI*	AAGTCGACCTAATAACGTGCACGCCGTGGCGA	

## Data Availability

The coordinates of the predicted models are available upon request to the corresponding author L.V.

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
