# Peer review of "Female Sex Determination Factors in Ceratitis capitata: Molecular and Structural Basis of TRA and TRA2 Recognition"

_insects, 2023, doi:10.3390/insects14070605_

Round 1

Reviewer 1 Report

In this study, the authors show by a yeast two-hybrid approach that the splicing factor TRA interacts molecularly with TRA2, and TRA2 with itself, in Ceratitis capitata. This is strictly similar to what is known in Drosophila melanogaster. In 2005, the same team had already made functional experiments by genetic means in drosophila showing that cctra can replace Drosophila tra in female sex determination.

Then, they provide Y2H experiments showing that the male-determining determinant (MoY) does not physically interact with either TRAF or TRA2, suggesting that MOY has no direct interaction with these proteins.

Finally, they use Alphafold to predict structures of the individual proteins CcTRA and CcTRA2 with limited new findings according to me. Then, they predict the structures of TRA-TRA2 and the ccTRA-DmTRA2 complexes to conclude that as expected, they share the same structural determinants.

In my opinion, this study does not bring a new knowledge in the field. The direct interaction between TRA and TRA2 protein is already well documented in other species and their previous paper was already showing that this is functionally true in Ceratitis capitata. Remaining experiments are computerized structural predictions with limited novelty.

Author Response

General comment

In my opinion, this study does not bring a new knowledge in the field. The direct interaction between TRA and TRA2 protein is already well documented in other species and their previous paper was already showing that this is functionally true in Ceratitis capitata. Remaining experiments are computerized structural predictions with limited novelty.

Response

We thank the reviewer for the evaluation of the manuscript. However, we believe that he/she overlooked our findings. Indeed, direct interaction of TRA and TRA2 has never been reported in Ceratitis capitata. Moreover, to our knowledge, this interaction has never been reported in other non-Drosophilidae insect species.

Reviewer 2 Report

Overall the manuscript by Perrotta et al. is quite interesting. Authors performed yeast two-hybrid (Y2H) assays to demonstrate the direct binding of the C. capitata TRA and TRA2. In addition, authors used machine-learning tools to predict protein structures and created a model of the CcTRA/CcTRA2 complex.

However, before publishing this manuscript the authors need to clarify the following text sections and perform the suggested experiment:

1) Line 33-34 “despite its highly high amino acid divergence” – remove highly

2) Y2H results indicate that when CcTRA-BD was baited with CcTRA2-AD as prey showed interaction. In contrast, no interaction occurred when CcTRA2-BD was used as bait and CcTRA-AD was used as prey.

Authors failed to explain and discuss why no interaction was observed when CcTRA2-BD was used as bait.    

3) Experiment suggestion: By altering the predicted interacting amino acids of the CcTRA and CcTRA2 Complex and performing Y2H assays, the predicted protein-protein interaction can be tested. This experiment will strengthen the manuscript.

Author Response

General comment

Overall the manuscript by Perrotta et al. is quite interesting. Authors performed yeast two-hybrid (Y2H) assays to demonstrate the direct binding of the C. capitata TRA and TRA2. In addition, authors used machine-learning tools to predict protein structures and created a model of the CcTRA/CcTRA2 complex.

Response

We thank the reviewer for his/her positive evaluation of the manuscript

Other points

1) Line 33-34 “despite its highly high amino acid divergence” – remove highly

Response

Done. We thank the reviewer for pointing out this typo

2) Y2H results indicate that when CcTRA-BD was baited with CcTRA2-AD as prey showed interaction. In contrast, no interaction occurred when CcTRA2-BD was used as bait and CcTRA-AD was used as prey. Authors failed to explain and discuss why no interaction was observed when CcTRA2-BD was used as bait.   

Response

In our Y2H experiment, the interaction between CcTRA-2 and CcTRA is observed when CcTRA2 is fused with the GAL4 AD and CcTRA with the BD, but not in the reciprocal combination. This result is possible because steric constraints or improper folding can occur in specific recombinant protein/GAL4 domain combinations. This issue has been previously reported when applying the Y2H analysis, and some examples of the lack of interaction between two interactors in both mutual directions can be found in the following papers:

 [1] Jia Y, McAdams SA, Bryan GT, Hershey HP, Valent B. Direct interaction of resistance gene and avirulence gene products confers rice blast resistance. EMBO J. 2000 Aug 1;19(15):4004-14. doi: 10.1093/emboj/19.15.4004. PMID: 10921881; PMCID: PMC306585.[2] Bogdanove AJ. Protein-protein interactions in pathogen recognition by plants. Plant Mol Biol. 2002 Dec;50(6):981-9. doi: 10.1023/a:1021263027600. PMID: 12516866. [3] Jiang X, Lubini G, Hernandes-Lopes J, Rijnsburger K, Veltkamp V, de Maagd RA, Angenent GC, Bemer M. FRUITFULL-like genes regulate flowering time and inflorescence architecture in tomato. Plant Cell. 2022 Mar 4;34(3):1002-1019. doi: 10.1093/plcell/koab298. PMID: 34893888; PMCID: PMC8894982. [4] Gross T, Broholm S, Becker A. CRABS CLAW Acts as a Bifunctional Transcription Factor in Flower Development. Front Plant Sci. 2018 Jun 20;9:835. doi: 10.3389/fpls.2018.00835. PMID: 29973943; PMCID: PMC6019494.

This issue has been commented on in the revised version of the manuscript. These related references have been cited.

3) Experiment suggestion: By altering the predicted interacting amino acids of the CcTRA and CcTRA2 Complex and performing Y2H assays, the predicted protein-protein interaction can be tested. This experiment will strengthen the manuscript.

Response

We have performed the AlphaFold prediction suggested by the reviewer. In particular, we tested the ability of a CcTRA variant, in which the putative TRA2 recognition motif (xxx residues), was deleted. In line with our CcTRA-CcTRA2 interaction model, no significant binding of this variant to CcTRA2 was observed. These new analyses have been added to the revised version of the manuscript.

Reviewer 3 Report

Comments to authors:

insects-2370088 Female sex determination factors in Ceratitis capitata: molecular and structural basis of TRA and TRA2 recognition

In this manuscript, Maryanna Martina et al. combined Y2H and machine-learning to explore the interaction between Tra and Tra2 in Ceratitis capitata. The authors found that CcTRA2 interacts with itself and generated a three-dimensional model of the complex formed by CcTRA and CcTRA2. The authors provide a new combined approach and an example to study protein interaction, even crossing species. The manuscript is written concisely and will be interesting for scientists in the insect field. I recommend acceptation of the manuscript with minor revision. 

Minor comments:

In the abstract, many words need italic: Drosophila melanogasterdoublesex (dsx) and fruitless (fru)…

Why the authors studied the interaction between MOY and CcTRA/CcTRA2? The MOY should be as the upstream signalling of CcTRA/CcTRA2.

The authors should provide the predicted structure of the CcTRA2 and DmTRA complex in the last part.

Author Response

General comment

In this manuscript, Maryanna Martina et al. combined Y2H and machine-learning to explore the interaction between Tra and Tra2 in Ceratitis capitata. The authors found that CcTRA2 interacts with itself and generated a three-dimensional model of the complex formed by CcTRA and CcTRA2. The authors provide a new combined approach and an example to study protein interaction, even crossing species. The manuscript is written concisely and will be interesting for scientists in the insect field. I recommend acceptation of the manuscript with minor revision.

Response

We thank the reviewer for his/her positive evaluation of the manuscript

Minor comments:

In the abstract, many words need italic: Drosophila melanogaster, doublesex (dsx) and fruitless (fru)…

Response

Done

Why the authors studied the interaction between MOY and CcTRA/CcTRA2? The MOY should be as the upstream signaling of CcTRA/CcTRA2.

Response

One hypothesis of MOY action is to inhibit the CcTRA/CcTRA2 complex in XY by interacting with one of the two proteins.

The authors should provide the predicted structure of the CcTRA2 and DmTRA complex in the last part.

Response

We have studied in silico only DmTRA2 and CcTRA complex but not the reciprocal combination because we connected this experiment with the data reported by Pane et al. (2005), which expressed CcTRA in Drosophila transgenic flies and suggested interaction between CcTRA and DmTRA2. A first attempt to produce Ceratitis transgenic flies expressing DmTRA failed because the PhD student (in 2003-2004) left for USA and the project was discontinued. Our main aim to confirm that CcTRA and CcTRA physically interact has been fulfilled.

Reviewer 4 Report

In the manuscript " Female sex determination factors in Ceratitis capitata: molecular and structural basis of TRA and TRA2 recognition ", Perrotta et al present a study in which they examined the interaction between TRA and TRA2 proteins in Ceratitis capitate using Y2H. Moreover, they generated structure prediction of the TRA and TRA2 complex using AlphaFold, and identified possible conserved domain in TRA that may be important for the interaction. I have some concerns as outlined below:

1. The authors found protein interaction when using bait CcTRA-BD with CcTRA2-AD as prey (Fig. 1, row 7), but not when using bait CcTRA2-BD with CcTRA-AD prey (Fig. 1, row 8). If there is interaction between TRA and TRA2, why does it only show in one condition and not the other? Can the authors give more explanation of the result?

2. The authors did not detect interaction between MOY and CcTRA/CcTRA2 in Y2H. To help better interpret this negative result, did the authors do western to confirm that the proteins are actually correctly expressed by the plasmid?

Minor points:

1. In line 189, it should be “To experimentally assess the direct interaction between CcTRA and CcTRA2”

2. In table 1, and figure 1/2 legend, TRA was labeled as TRAF. What is the meaning of the “F”?

Overall, the manuscript is well written and clear, and easy to follow.

Author Response

General comment

In the manuscript " Female sex determination factors in Ceratitis capitata: molecular and structural basis of TRA and TRA2 recognition ", Perrotta et al present a study in which they examined the interaction between TRA and TRA2 proteins in Ceratitis capitata using Y2H. Moreover, they generated structure prediction of the TRA and TRA2 complex using AlphaFold, and identified possible conserved domain in TRA that may be important for the interaction.

Response

We thank the reviewer for the further suggestions and following concerns.

I have some concerns as outlined below:

  1. The authors found protein interaction when using bait CcTRA-BD with CcTRA2-AD as prey (Fig. 1, row 7), but not when using bait CcTRA2-BD with CcTRA-AD prey (Fig. 1, row 8). If there is interaction between TRA and TRA2, why does it only show in one condition and not the other? Can the authors give more explanation of the result?

Response

In our Y2H experiment, the interaction between CcTRA-2 and CcTRA is observed when CcTRA2 is fused with the GAL4 AD and CcTRA with the BD, but not in the reciprocal combination. This result is possible because steric constraints or improper folding can occur in specific recombinant protein/GAL4 domain combinations. This issue has been previously reported when applying the Y2H analysis, and some examples of the lack of interaction between two interactors in both mutual directions can be found in the following papers:

Jia Y, McAdams SA, Bryan GT, Hershey HP, Valent B. Direct interaction of resistance gene and avirulence gene products confers rice blast resistance. EMBO J. 2000 Aug 1;19(15):4004-14. doi: 10.1093/emboj/19.15.4004. PMID: 10921881; PMCID: PMC306585.

Bogdanove AJ. Protein-protein interactions in pathogen recognition by plants. Plant Mol Biol. 2002 Dec;50(6):981-9. doi: 10.1023/a:1021263027600. PMID: 12516866.

Jiang X, Lubini G, Hernandes-Lopes J, Rijnsburger K, Veltkamp V, de Maagd RA, Angenent GC, Bemer M. FRUITFULL-like genes regulate flowering time and inflorescence architecture in tomato. Plant Cell. 2022 Mar 4;34(3):1002-1019. doi: 10.1093/plcell/koab298. PMID: 34893888; PMCID: PMC8894982.

Gross T, Broholm S, Becker A. CRABS CLAW Acts as a Bifunctional Transcription Factor in Flower Development. Front Plant Sci. 2018 Jun 20;9:835. doi: 10.3389/fpls.2018.00835. PMID: 29973943; PMCID: PMC6019494.

We have added text and references in the MS to clarify this point.

  1. The authors did not detect interaction between MOY and CcTRA/CcTRA2 in Y2H. To help better interpret this negative result, did the authors do western to confirm that the proteins are actually correctly expressed by the plasmid?

Response

Unfortunately, we do not have a western blot showing that MOY fusion protein was actually expressed in the yeast. However, these experiments with MOY were performed in the same experimental conditions in which CcTRA-CcTRA-2 and CcTRA2-CcTRA2 interactions were detected, indicating their successful expression. All plasmid clones were sequenced. By inference, we do not expect that MOY fusion proteins are not expressed. But we agree that we do not have direct evidence. It will take too long to get these control data to support a negative result which has already its own limitations as we discussed in the conclusions.

Minor points:

  1. In line 189, it should be “To experimentally assess the direct interaction between CcTRA and CcTRA2”

Response

Thank you. DONE.

  1. In table 1, and figure 1/2 legend, TRA was labeled as TRAF. What is the meaning of the “F”?
  2.  

Response

We corrected CcTRAF in CcTRA.

Round 2

Reviewer 1 Report

The revised version of this article has not addressed any of the points I raised during my first assessment of this work. Lack of novelty was my only concern and I cited a specific work performed in 2005 by the same group where they already concluded that ccTra and Tra2 can interact and constitute a functional complex. According to me, an additional 2 hybrid experiment in yeast showing an interaction does not bring any new knowledge in the field.

Author Response

Response

We thank the reviewer for the evaluation of the revised manuscript. In his/her evalutation of the original version of the MS, the only point raised was referred to the novelty of the data presented. As already underlined in the first rebuttal round, in Pane et al. (2005) transgenic flies of D. melanogaster expressing CcTRA protein were used only to rescue the endogenous tra mutation in the presence of a Dmtra-2 endogenous gene. These findings were suggesting that CcTRA could interact with DmTRA2 directly. No additional experiments were provided suggesting that CcTRA could interact with CcTRA2. The question if CcTRA interact with CcTRA2 was left open at that time and until now. Therefore, experimental data showing the direct interaction between TRA and TRA2 had never been reported in Ceratitis capitata. Moreover, to our knowledge, this interaction has never been reported in other non-Drosophilidae insect species.

As we stated in the conclusions, “collectively the data here presented show that the Y2H approach, combined with powerful predictive tools of three-dimensional protein structures, could use helpful also in other insect species to understand the potential links between different proteins acting as primary sex-determining signals and the conserved TRA and TRA2 transducers.”.

Reviewer 2 Report

The authors address all of my comments.

Minor comment:

Citation missing in line 366

Author Response

We thank the reviewer 2 for his/her re-evaluation of the MS. We added the missing references in line 366.

Reviewer 4 Report

In the revised manuscript, the authors added text to address my concerns. They also answered most of my concerns in the rebuttal letter.

Author Response

We thank the reviewer 4 for his/her re-evaluation of the MS.